# Risk factors and risk profiles for neck pain in young adults: Prospective analyses from adolescence to young adulthood—The North-Trøndelag Health Study

Henriette Jahre[1]*, Margreth Grotle[1,2], Milada Småstuen[3], Maren Hjelle Guddal[2], Kaja Smedbråten[1], Kåre Rønn Richardsen[1], Synne Stensland[2,4], Kjersti Storheim[1,2], Britt Elin Øiestad[1]

1 Department of Physiotherapy, Oslo Metropolitan University, Oslo, Norway, 2 Research and Communication Unit for Musculoskeletal Health (FORMI), Clinic for Surgery and Neurology, Oslo University Hospital, Oslo, Norway, 3 Department of Nursing, Oslo Metropolitan University, Oslo, Norway, 4 Norwegian Centre for Violence and Traumatic Stress Studies (NKVTS), Oslo, Norway

* henriett@oslomet.no

**Data Availability Statement:** Data cannot be shared publicly because of restrictions from the Regional Committees for Medical and Health Research Ethics (post@helseforskning.etikkom.no)

## Abstract

The objective was to investigate risk factors and risk profiles associated with neck pain in young adults using longitudinal data from the North-Trøndelag Health Study (HUNT). Risk factors were collected from adolescents (13–19 years of age), and neck pain was measured 11 years later. The sample was divided into two: Sample I included all participants (n = 1433), and Sample II (n = 832) included only participants who reported no neck/shoulder pain in adolescence. In multiple regression analyses in Sample I, female sex (OR = 1.9, 95% CI [1.3–2.9]), low physical activity level (OR = 1.6, 95% CI [1.0–2.5]), loneliness (OR = 2.0, 95% CI [1.2–3.5]), headache/migraine (OR = 1.7, 95% CI [1.2–2.6]), back pain (OR = 1.5, 95% CI [1.0–2.4]) and neck/shoulder pain (OR = 2.0, 95% [CI 1.3–3.0]) were associated with neck pain at the 11-year follow-up. Those with a risk profile including all these risk factors had the highest probability of neck pain of 67% in girls and 50% in boys. In Sample II, multiple regression analyses revealed that female sex (OR = 2.2, 95% CI [1.3–3.7]) and perceived low family income (OR = 2.4, 95% CI [1.1–5.1]) were associated with neck pain at the 11-year follow-up. Girls and boys with a perceived low family income had a 29% and 17% higher probability of neck pain than adolescents with a perceived high family income. The risk profiles in both samples showed that co-occurrence of risk factors, such as headache/migraine, neck/shoulder pain, back pain, low physical activity level, loneliness, and perceived low family income cumulatively increased the probability of neck pain in young adulthood. These results underline the importance of taking a broad perspective when studying, treating, and preventing neck pain in adolescents.

in accordance with Norwegian law, as participants in the HUNT survey have not given consent to the public sharing of their data. Therefor, these data are only available upon appropriate request to the Data Access Committee at HUNT Research Centre at hunt@medisin.ntnu.no.

**Funding:** Oslo Metropolitan University funded this paper to HJ through a PhD position.

**Competing interests:** The authors have declared that no competing interest exists.

## Introduction

According to the Global Burden of Disease Study [1], neck pain is one of the most common musculoskeletal (MSK) disorders worldwide and is a top-five cause of years lived with disability in high and middle-income countries. Neck pain is reported as the most prevalent MSK pain site among adolescents [2, 3], commonly accompanied by low health-related quality of life, school absence, and avoidance of participation in activities and sports [3, 4]. Importantly, neck pain become persistent in many adolescents, but there is little knowledge regarding risk factors and causes of neck pain in these individuals [5]. The high prevalence of neck pain in adolescents [2, 6] is of great concern since studies suggest that individuals who develop pain and disabilities during adolescence are more likely to report these health complaints in adulthood [7–9].

Adolescence is a period of life characterised by significant changes in both the biology and the social environment. Development of MSK pain may be influenced by such changes and characteristics, for instance, sleeping disturbances [10], mental health problems [11], and a decreased physical activity level [12]. Social factors have been less studied, but peer-related stress [13] and loneliness [2] have shown associations with MSK pain in longitudinal studies of adolescents. We systematically reviewed longitudinal studies investigating risk factors for neck pain in young adults (18–29 years old). The searches revealed six studies analysing more than 50 risk factors, however no consistent risk factors were identified [14]. Cross-sectional studies have shown significant associations between neck pain and psychosocial factors, such as anxiety, depression, and perceived stress [15, 16]. Studies of adults indicate that risk factors for neck pain often encompassing a range of biopsychosocial factors [17] and apparently, there are differences depending on the inclusion of participants with a presence or an absence of neck pain at baseline [17]. Identifying single risk factors and the co-occurrence of risk factors in adolescents will enable us to identify high-risk groups. Such knowledge will contribute to designing future preventive interventions with the aim of reducing the high burden of neck pain for both the society and the individuals affected. To the best of our knowledge, no previous studies have investigated co-occurrence of risk factors (risk profiles) for neck pain in adolescents.

The objective of this study was to investigate the associations between potential risk factors and risk profiles in adolescence and neck pain in young adulthood in an 11-year prospective population-based study.

## Materials and methods

The current study used data from the North-Trøndelag Health Study (HUNT), a large prospective population-based cohort study conducted in North-Trøndelag County in Norway [18, 19]. The HUNT Study consists of four health surveys carried out with 11-year intervals, where all inhabitants of North-Trøndelag County above 13 years of age were invited to participate. The surveys are divided into Young-HUNT (13–19 years of age) [19] and HUNT (20 years of age and above) [18]. To answer our research questions, we have linked data from wave 3 and 4, i.e. Young-HUNT3 (2006–2008) and HUNT4 (2017–2019).

Participation in the HUNT study was voluntary, and all study participants signed a written consent form. Written consent from a guardian was required for participants under the age of 16 years. The Regional Committee for Medical and Health Research Ethics (2019/517/REK Midt) and the Norwegian Centre for Research Data (543422) approved the present study. The study protocol has been published at clinical-trials.gov (NCT04201366). Reporting of this study is following the Strengthening the Reporting of Observational Studies in Epidemiology (STROBE) guidelines [20] (S1 Table).

## Study population

Adolescents between 13 to 19 years of age from the Young-HUNT3 Study were included. To investigate both first-onset neck pain and new episodes of neck pain, participants were divided into two samples. Sample I includes all participants with valid data from both time points (Young-HUNT3 and HUNT4 Studies). Sample II includes only those who reported *"never/ seldom"* to neck/shoulder pain in Young-HUNT3 (pain-free at baseline). Individuals who reported juvenile arthritis were excluded (Fig 1). Of the 8122 adolescents who were included in the Young-HUNT3 Study, 1433 (17.6%) attended the HUNT4 Study (Sample I). In Sample II (those who were pain-free at baseline), 823 (11.0%) fulfilled the inclusion criteria, responded to follow-up, and were included in the analyses in this study. A considerable number of participants were not re-invited in HUNT4 because they moved from the Municipality, moved to another country, or died during follow-up (n = 2931 (36%)). Fig 1 illustrates the flow-chart of the study participants.

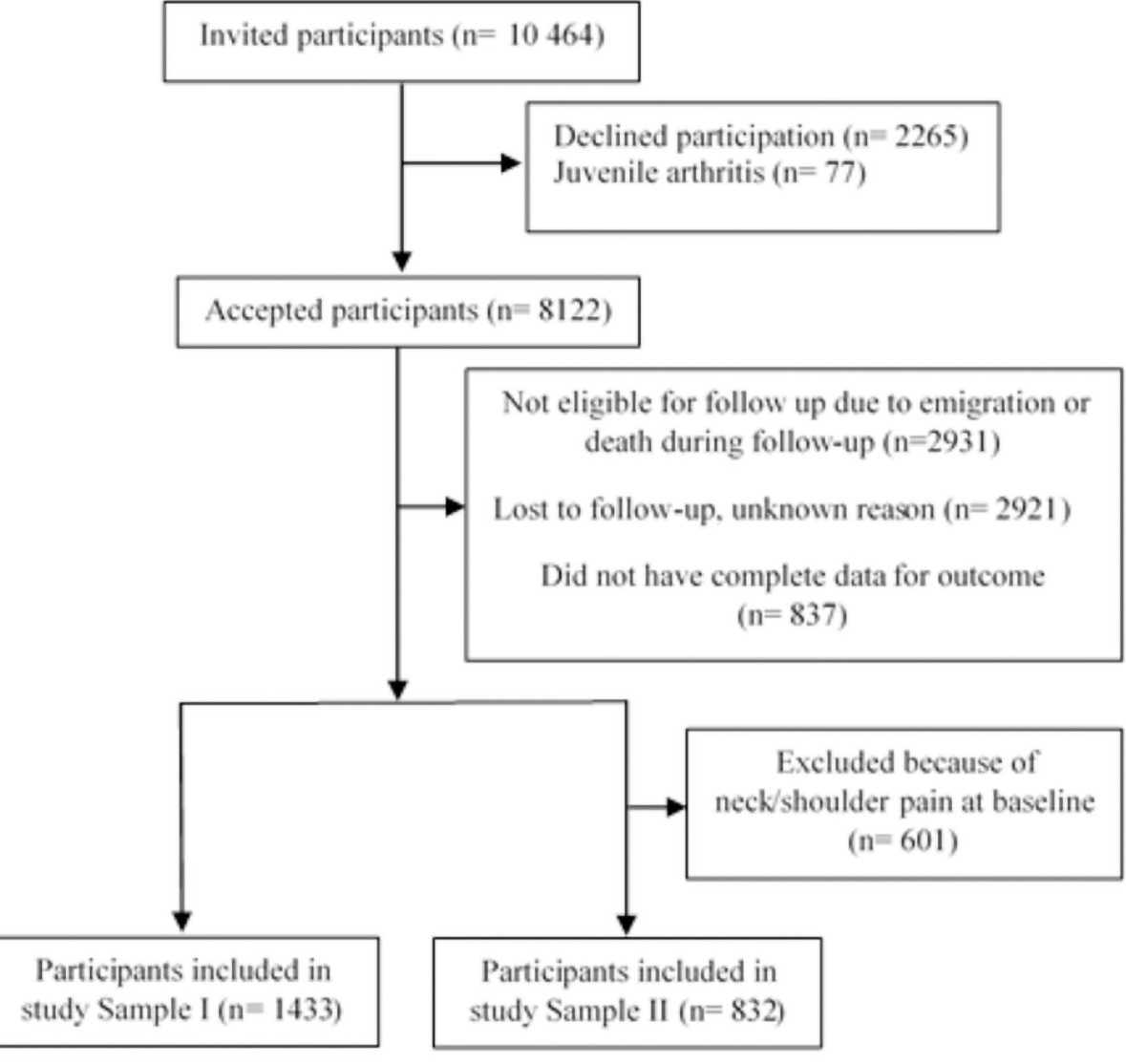

**Fig 1. Flow-chart of study participants.**

## Procedure

The Young-HUNT3 data collection took place during school hours with a comprehensive questionnaire, physical tests, and measurements of height and weight. Adolescents not attending the school on the day of assessment received the questionnaire by mail. They were invited to do the physical examinations at a field station in the local area. The HUNT4 Study took place 11 years later. The participants received an invitation letter and were asked to complete the electronic questionnaire and undertake physical tests at a local field station.

## Potential risk factors (Young-HUNT3)

Investigated risk factors were chosen based on previous research of neck and other MSK pain in adolescents and adults. Neck pain seems to have multifactorial causation consisting of a range of biopsychosocial factors, such as anthropometric factors, previous pain conditions [21–23], lifestyle factors [10], psychological factors [24], and social factors [2, 25].

**Anthropometric factors.**   Height and weight were objectively measured by a trained nurse. Body mass index (BMI) was calculated as $kg/m^2$. Age-adjusted cut-offs from Cole et al. (2012) defined the BMI categories: *"thinness"*, *"normal weight"*, and *"overweight/obese"* [26]. Due to the low number of participants located in the *thinness* group (4%), *thinness* and *normal weight* were merged into one category.

**Pain.**   In Young-HUNT3, pain was measured with items developed by Mikkelson et al., which have shown good concurrent validity and test-retest reliability in adolescents [27]. Participants were asked: *"How often have you had any of the below-listed pain during the last three months?"* A body chart of 11 body regions accompanied the question. The body regions were *headache/migraine*, *neck/shoulder pain*, *back pain* (upper and lower back), *chest pain*, *upper extremity pain* (left and right arm), *lower extremity pain* (left and right leg), *abdominal pain*, and *other pain*. For each body region, response categories were on a five-point Likert scale from "*never/seldom*" to "*almost every day*". A cut-off score of more than one day per week was set to distinguish between participants with frequent pain and those with infrequent pain. Pain in each specific region was investigated as single risk factors.

To investigate if several pain sites in adolescence was a risk factor for neck pain in young adulthood, we created the variable *number of pain sites* by summarising all the 11 body regions. The number of pain sites was categorised into "*no pain*", "*one pain site*", "*two pain sites*" and "*three or more pain sites*" due to a low number of participants having more than three pain sites.

**Lifestyle factors.**   Sleep problems, defined as having difficulty falling asleep at night, was measured on a four-point Likert scale ranging from "*almost every night*" to "*never*". This variable was recoded into "*never*", "*seldom*" or "*almost every night*". This question is made by the Norwegian Institute of Public Health, inspired by similar questions from other health studies, but is not formally validated.

Physical activity was assessed with a question adapted from the World Health Organization Health Behaviour in Schoolchildren (HBSC) study [28]. The question has showed to correlate with cardiovascular fitness (r = 0.39), especially for girls (r = 0.55) [29]. Participants were asked how many days a week outside school hours they play sports or exercise to the point where they breathe heavily and/or sweat. This question had seven response categories ranging from "*never*" to "*every day*". The response alternatives were operationalised into three levels: "*one day a week or less (low level)*", "*two to three days a week (moderate level)*", and "*four days a week or more (high level)*" as in a previous study [30].

**Psychological factors.**   A five-item short version of the Symptoms Checklist (SCL-5) was used to measure symptoms of psychological distress, which is validated in Norwegian

adolescents [31, 32]. This questionnaire consisted of five questions measuring whether the adolescents had been bothered with feelings of "*fear or anxiety*", "*tension or restlessness*", "*hopelessness about the future*", "*dejection or sadness*", and/or "*excessive worry*". These checklist items were scored using a four-point Likert scale ranging from "*not troubled*" to "*very much troubled*", referring to symptoms the previous two weeks. Higher scores indicate a higher level of psychological distress. The SCL-5 has demonstrated high reliability and high correlation with SCL-25 and SCL-10 [32], which have been validated in adolescents [33]. A cut-off score of $\geq 2.0$ defined the presence of psychological distress, as suggested in one study [32].

Self-esteem was measured with four questions from the Rosenberg self-esteem scale (RSE) [34] and used as a continuous variable in the analyses. Each question was scored on a four-point scale ranging from "*strongly agree*" to "*strongly disagree*". The short version of the RSE has shown a high correlation with the full version [35], which has demonstrated good validity in adolescents [34].

**Social factors.** Resilience was measured with eight questions from the Resilience Scale for Adolescents (READ) [36]. READ has shown acceptable validity in Norwegian adolescents [37]. The subscales *social competence* and *family cohesion* from the original questionnaire were used based on recommendations from the original developers. Social competence included four questions regarding their ability to: *make other people feel comfortable around them*, *find new friends*, *talk to new people*, and *find something fun to talk about*. Family cohesion included questions regarding *shared family values*, *well-being within the family*, *shared positive expectations*, and *support of each other*. Each question was rated on a five- points Likert scale ranging from "*I totally agree*" to "*I totally disagree*". A higher score indicates high resilience.

Loneliness was measured with the question: "*do you often feel lonely*?" with five response alternatives that were transformed into three categories: "*often/very often*", "*sometimes*", "*seldom or never*". This one-item question has been employed in one study measuring loneliness in adolescents [2], but is not formally validated.

The perceived family income was measured with one question from the HBSC study [28] asking: "*How well off do you think your family is compared to most others*?" The responses were: "*about the same as most others*", "*better financial situation*", and "*worse financial situation*". This question has shown correspondence with parents' education and parents' work affiliation in a previous Norwegian study [38], but is not formally validated.

## Outcome measures (HUNT4)

The primary outcome was neck pain measured at the 11-year follow-up (young adulthood). Neck pain was defined when lasting for three consecutive months or more during the last year and based on the question: "*In the last year, have you had pain or stiffness in muscles or joints that has lasted at least three consecutive months*?" The responses were *yes* or *no*. If participants answered *yes*, they were asked "*where have you had this pain or stiffness*", accompanied by a body chart divided into different body regions. Participants who answered *yes* to pain and marked on *neck* in the chart were identified as cases in this study.

## Statistical analyses

Continuous variables describing the study samples were reported with means and standard deviations (SD) when normally distributed and medians and ranges if they had skewed distributions. Categorical variables were reported as counts and percentages. Bivariate analyses of baseline characteristics were conducted to investigate possible differences between responders and non-responders. The chi-square test was used to compare categorical variables,

independent sample t-test for normally distributed continuous variables, and Mann Whitney U test for pairs of data with skewed distribution.

We used univariate binary logistic regression analyses to analyse crude associations between each potential risk factor and neck pain. Variables with a p-value ≤ 0.1 in these univariate analyses were included in a multiple model using a backward stepwise selection [39] (S2 Table). P-values ≤ 0.05 were considered statistically significant in the multiple regression models. The results were expressed as odds ratios (ORs) with 95% confidence intervals (CIs). Assessment of collinearity was conducted before the inclusion of variables in the multiple models. Missing data on potential risk factors was first handled by multiple imputations. The univariate analyses with imputed data showed similar results as the complete case analyses presented in the paper.

Risk profiles for neck pain in young adulthood were identified by converting the coefficients from the multiple regression analyses into probabilities given different combinations of significant risk factors using the following formula [40]:

$$P(\text{Y} = 1|\mathbf{X}) = \frac{e^{b_0 + b_1 x_1 + b_2 x_2 + b_3 x_3 + b_4 x_4 + b_5 x_5 + b_6 x_6}}{1 + e^{b_0 + b_1 x_1 + b_2 x_2 + b_3 x_3 + b_4 x_4 + b_5 x_5 + b_6 x_6}}$$

where $b_0$, $b_1 x_1$, $b_2 x_2$, $b_3 x_3$, $b_4 x_4$, $b_5 x_5$, $b_6 x_6$ were the significant risk factors from the final multiple regression model.

Risk matrices were used to visualise the results, as reported in previous studies [41, 42]. The matrices are presented separately for girls and boys. All analyses were conducted using SPSS statistical software (SPSS Inc, Chicago, IL, USA).

## Results

### Demographics of study participants

Baseline characteristics are presented in Table 1. The mean age was 16 years (SD 1.8), and there was a higher proportion of girls (63% in Sample I and 57% in Sample II). Most study participants reported moderate to high physical activity level and a BMI within a normal weight range. The prevalence of neck/shoulder pain in all study participants (Sample I) was 18.1% (95% CI [16–20]). Headache/migraine was the most prevalent pain condition. The proportions of missing values for risk factors ranged from 0.9% to 13%.

Analyses comparing our Sample I with those lost to follow-up showed that statistically significantly more males than females were lost to follow-up (52% vs 47%), the non-responders had a higher baseline physical activity level, and higher baseline self-esteem compared to responders (S3 Table).

### Neck pain in young adulthood

At follow-up, 18.4% (95% CI [16–20]) of all respondents reported neck pain (Sample I). Among the pain-free adolescence at baseline (Sample II), 12.1% (95%CI [10–14]) reported neck pain at follow-up. Thirty-six percent of those with neck/shoulder pain at baseline had neck pain at follow-up.

### Risk factors for neck pain in all study participants (Sample I)

In univariate analyses, female sex, high BMI, perceived low family income, headache/migraine, neck/shoulder pain, back pain, abdominal pain, three or more pain sites, low physical activity level, sleeping problems, psychological distress, low family cohesion, loneliness, and self-esteem were significantly associated with neck pain at the 11-year follow-up (S2 Table). The

**Table 1. Baseline characteristics of study participants.**

| Characteristics at baseline | Sample I | Sample II |
|---|---|---|
| | N = 1433 | N = 832 |
| Age (*yr*) *mean*, *(SD)* | 15.9 (1.8) | 15.8 (1.7) |
| Sex, female *n (%)* | 912 (63.6) | 479 (57.6) |
| Perceived family income, *n (%)* | | |
| Better | 215 (15.0) | 127 (15.3) |
| Average | 1006 (70.2) | 598 (71.9) |
| low | 118 (8.2) | 59 (7.1) |
| *Missing* | 94 (6.6) | 48 (5.8) |
| BMI *(kg/m²)*, *n (%)* | | |
| Normal | 1037 (72.4) | 621 (74.6) |
| Overweight/obese | 315 (22.0) | 161 (19.4) |
| *Missing* | 81 (5.7) | 50 (6.0) |
| School type, *n (%)* | | |
| Middle school students, *n* | 808 (56.4) | 489 (58.8) |
| High school students, *n* | 592 (41.3) | 326 (39.2) |
| *Missing* | 33 (2.3) | 17 (2.0) |
| Subjective health, *n (%)* | | |
| Good | 1283 (89.5) | 783 (94.1) |
| Poor | 137 (9.6) | 44 (5.3) |
| *Missing* | 13 (0.9) | 5 (0.6) |
| Neck/shoulder pain, *n (%)* | | |
| Often[¥] | 259 (18.1) | |
| *Missing* | 25 (1.7) | |
| Headache/migraine, *n (%)* | | |
| Often[¥] | 339 (23.7) | 118 (14.2) |
| *Missing* | 26 (1.8) | 6 (0.7) |
| Back pain, *n (%)* | | |
| Often[¥] | 266 (18.6) | 58 (7.0) |
| *Missing* | 40 (2.8) | 10 (1.2) |
| Number of pain sites, *n (%)* | | |
| 0 | 668 (46.6) | 536 (64.4) |
| 1 | 237 (16.5) | 142 (17.1) |
| 2 | 142 (9.9) | 57 (6.9) |
| 3 or more | 213 (14.9) | 22 (2.6) |
| *Missing* | 173 (12.1) | 75 (9.0) |
| Physical activity level, *n (%)* | | |
| High level | 526 (36.7) | 329 (39.5) |
| Moderate level | 513 (35.8) | 299 (35.9) |
| Low level | 372 (26.0) | 190 (22.8) |
| *Missing* | 22 (1.5) | 14 (1.7) |
| Psychological distress[‡], *n (%)* | | |
| <2.00 | 1124 (78.4) | 730 (87.7) |
| ≥2.00 | 266 (18.6) | 80 (9.6) |
| *Missing* | 43 (3.0) | 22 (2.6) |
| Loneliness, *n (%)* | | |
| Often | 119 (8.3) | 41 (4.9) |
| Sometimes | 319 (22.3) | 152 (18.3) |

(*Continued*)

**Table 1.** (Continued)

| Characteristics at baseline | Sample I | Sample II |
|---|---|---|
| | N = 1433 | N = 832 |
| Seldom | 899 (62.7) | 586 (70.4) |
| *Missing* | 96 (6.7) | 53 (6.4) |
| Self-esteem[†], *mean (SD)* | 8.0 (2.4) | 8.4 (2.3) |
| *Missing* | 77 (5.4) | 42 (5.0) |

Sample I = all participants, Sample II = pain-free participants at baseline.

yr, year; SD, standard deviation; BMI, body mass index.

[¥] Pain at least once per week during the last three months not related to any known disease or injury [‡] Symptom check list (1–4), [†] Rosenberg self-esteem scale (0–12).

multiple logistic regression analyses showed that female sex, headache/migraine, neck/shoulder pain, back pain, low physical activity level, and loneliness were all independently statistically significantly associated with neck pain at the 11-year follow-up (Table 2).

### Risk factors for neck pain in the study participants pain-free at baseline (Sample II)

When assessing crude associations between potential risk factors and neck pain among those who were pain-free at baseline, our data revealed higher odds for neck pain for female sex, perceived low family income, headache/migraine, three or more pain sites, low physical activity level, sleeping problems, loneliness, and self-esteem (S2 Table). In the multiple logistic regression analyses, we found that female sex and perceived low family income remained significantly associated with neck pain at the follow-up (Table 2).

### Risk profiles

**Sample I.** The highest probability of neck pain was found in participants with a low level of physical activity, loneliness, headache/migraine, back pain, and neck/shoulder pain (Fig 2). In these participants, the probability of having neck pain at follow-up was 67% (95% CI [65–70]) among the girls and 50% (95% CI [47–53]) among the boys. This was compared to 13% (95% CI [12–15]) among the girls and 7% (95% CI [6–8]) among the boys with moderate to high physical activity level, not feeling lonely and no headache/migraine, back pain or neck/shoulder pain.

**Sample II.** Fig 3 displays the risk profiles for neck pain in a visual risk matrix for those who were pain-free at baseline (Sample II). The probability of having neck pain at the 11-year follow-up was 29% (95% CI [26–32]) among the girls with perceived low family income and 17% (95% CI [14–20]) among the boys with perceived low family income.

## Discussion

This study found that female sex, low level of physical activity, loneliness, headache/migraine, back pain, and neck/shoulder pain in adolescence were risk factors for having neck pain in young adulthood in Norwegian adolescents. The co-occurrence of these risk factors during adolescence cumulatively increased the probability of neck pain in young adulthood. Among adolescents without neck/shoulder pain at baseline, significant risk factors were female sex and perceived low family income.

**Table 2. Multiple analysis of the association between potential risk factors in adolescence and persistent neck pain in young adulthood.**

| | Association with neck pain | |
| --- | --- | --- |
| | Sample I (n = 1433) | Sample II (n = 832) |
| **Variables** | **OR and 95% CI** | **OR and 95% CI** |
| Sex | | |
| Male | 1.0 | 1.0 |
| Female | 1.9 (1.3–2.9)* | (1.3–3.7)* |
| Perceived family income | | |
| Average | | 1.0 |
| Better | | 1.3 (0.6–2.5) |
| Worse | | 2.4 (1.1–5.1)* |
| Headache/migraine | | |
| Seldom | 1.0 | |
| Often[¥] | 1.7 (1.2–2.6)* | |
| Neck/shoulder pain | | |
| Seldom | 1.0 | |
| Often[¥] | 2.0 (1.3–3.0)* | |
| Back pain | | |
| Seldom | 1.0 | |
| Often[¥] | 1.5 (1.0–2.4)* | |
| Physical activity | | |
| High level | 1.2 (0.8–1.8) | |
| Moderate level | 1.0 | |
| Low level | 1.6 (1.0–2.5)* | |
| Loneliness | | |
| Seldom | 1.0 | |
| Sometimes | 1.2 (0.9–2.0) | |
| Often/very often | 2.0 (1.2–3.5)* | |

Sample I = all participants, Sample II = pain-free participants at baseline.

All significant variables were adjusted for each other.

1.0 = reference category.

*Variables significantly associated with persistent neck pain.

[¥] Pain at least once per week during the last three months not related to any known disease or injury.

Our finding that participants with combinations of risk factors in adolescence had a cumulatively increased probability of neck pain in young adulthood is in line with a previous cross-sectional study investigating the relationship between lifestyle behaviour and chronic non-specific pain in Norwegian adolescents [43]. They found a gradually stronger association with a higher number of unhealthy variables (low physical activity level, sedentary behaviour, high BMI, smoking, and alcohol). As illustrated in the risk matrix, the highest probability of neck pain occurred when combining all the statistically significant variables from the multiple analyses, and the probability decreased when individuals had more favourable outcomes. One might speculate that it is the actual number of risk factors that increase the probability of neck pain, regardless of type. Girls had, in general, a higher probability of neck pain than boys regardless of the combinations of risk factors.

Previous systematic reviews of children, adolescents, and young adults have shown inconsistent results for sex as a risk factor for MSK pain [11, 14]. One systematic review found

|  |  | Girls | | Boys | |  |
|---|---|---|---|---|---|---|
|  |  | **No loneliness** | **Loneliness** | **No loneliness** | **Loneliness** |  |
| **Moderate/ high physical activity level** | **No neck/shoulder pain** | 13% (12-15) | 31% 28-33) | 7% (6-8) | 16% (14-18) | **No Headache/migraine** |
|  | **Neck/shoulder pain** | 35% (32-37) | 58% (55-60) | 21% (19-23) | 40% (37-42) | **Headache/migraine** |
| **Low physical activity level** | **No neck/shoulder pain** | 19% (17-21) | 37% (35-40) | 10% (9-12) | 22% (20-25) | **No headache/migraine** |
|  | **Neck/shoulder pain** | 45% (42-47) | 67% (65-70) | 28% (26-31) | 50% (47-53) | **Headache/migraine** |
|  |  | **No Back pain** | **Back pain** | **No back pain** | **Back pain** |  |

**Fig 2. Risk profiles for persistent neck pain in young adulthood in Sample I (n = 1433).** Sample I = all participants. Probabilities of persistent neck pain at follow-up (%, [95% CI]), red = highest risk profile.

inconsistent results across three studies investigating sex as a risk factor for neck pain in young adults [14], and a meta-analysis of Huguet et al. found inconsistent results of sex as a risk factor for MSK pain in children and adolescents [11]. Huguet et al., however, identified in their subgroup analyses that these differences might be caused by different pain conditions (chronic, acute, or mixed) included in the studies. They found that clearly defined chronic/recurrent MSK pain was significantly associated with female sex. This is in line with our finding of sex as a risk factor for neck pain lasting three months or more. Many explanations have been proposed for sex differences, such as innate differences in visceral and somatic perception, lower pain threshold in girls, and differences in reporting and acknowledgement of discomfort [44].

| | Girls | Boys |
|---|---|---|
| **Average/better family economy** | 13% (10-15) | 7% (5-8) |
| **Low family economy** | 29% (26-32) | 17% (14-20) |

**Fig 3. Risk profiles for persistent neck pain in young adulthood in Sample II (n = 832).** Sample II = pain-free participants at baseline. Probabilities of persistent neck pain at follow-up (%, [95% CI]), red = highest risk profile.

Further, there are differences in physical growth and development, psychological maturation, and hormonal profile during adolescence. This might influence the reporting of neck pain [45].

Our finding that headache/migraine, neck/shoulder, and back pain in adolescence were associated with neck pain in young adulthood among Norwegian adolescents is supported in the literature [21, 46, 47]. The reasons for these associations are unclear, but changes caused by pain in one body site might influence other body sites, and share similar mechanisms. One explanation for pain in different body parts is the neurophysiological changes implicated in central sensitisation. Further, studies have shown alterations in pain processing after an episode of acute pain, which seems to influence pain persistence in adolescents [48]. Our finding of a high prevalence of neck/shoulder pain in adolescence and the impact early pain had on future pain indicate that pain develops early.

Contrary to previous studies [14, 21], we found that a low level of physical activity in adolescence was associated with neck pain in young adulthood (Sample I). Differences in measurement of physical activity and in activities conducted [49] may explain inconsistent findings. Another reason might be the possible fluctuations in physical activity level during the current study's long follow-up period [50]. Finally, we could speculate if adolescents with a low physical activity level do more sedentary activities such as screen-based activities. Screen-based activities have shown association with neck pain in one previous study [51].

Our finding that loneliness was associated with neck pain (Sample I) is in line with results from two Scandinavian cross-sectional studies investigating associations between loneliness related to spinal pain [2] and headache in adolescents [52]. Other social determinants, such as bullying and peer-related stress, have been associated with MSK pain in previous longitudinal studies [13, 53]. One explanation for these associations might be commonalities in the neurobiology seen in both "social pain" and physical pain. A study using functional magnetic resonance imaging found that social exclusion activated the same brain regions similar to physical pain [54]. Further, lonely adolescents could be more affected by negative emotions and be less able to cope with pain than adolescents with healthy social relationships. Studies of patients with chronic pain show that a lack of perceived social support is associated with higher pain intensity [55].

In line with other studies investigating associations between low socioeconomic status and economic stress in MSK pain in children, adolescents [11], and adults [56], we found an association between perceived low family income and neck pain in young adulthood (Sample II). However, this study's measurement of socioeconomic status is different than most other studies, including the adolescent's self-perceived family income rather than measuring socioeconomic status using parent's education or work status. It is essential to consider the cultural aspect of this question. In a country like Norway with a high welfare standard, perceived low family income may be more related to possibilities for social participation rather than poverty related to having access to water, food, or healthcare. Thus, our results might be more related to the social aspects of having a low family income, such as lower possibilities of social participation in different activities.

Analysis derived from Sample I and Sample II revealed different risk factors, except for sex. An explanation for these results can be that Sample II excluded those with neck/shoulder pain at baseline. Previous studies have shown that adolescents experiencing pain have other illness perceptions or health behaviours than those who are pain-free. This includes withdrawal from social [57, 58] and pain-provoking physical activities [3, 58], reduced sleep quality [58, 59], decreased quality of life [4], and lower psychosocial well-being [2]. These factors might influence future pain experience [60]. Another reason for different results between the two samples might be lack of statistical power due to the sample sizes (1422 vs 832). For instance, Sample II

included few cases who experienced loneliness (n = 41), back pain (n = 58), and headache/migraine (n = 118). Also, by excluding participants with a previous episode of neck pain, we probably excluded participants with other pain sites. This might explain lack of statistical significance for headache/migraine and back pain.

Psychological distress and sleeping difficulties did not reach statistical significance in our multiple analyses. This is contrary to findings from previous studies [10, 61]. Potential explanations may be different measurements used, different follow-up periods, and different statistical models.

## Strengths and limitations

The strengths of this study are the prospective design, and the large sample size at baseline. The novelty of this study is our analytic approach of combining risk factors in risk profiles, and arranging the probability of having neck pain in young adulthood for given combinations of risk factors using risk matrices. To our best knowledge, this is the first study investigating risk profiles for neck pain in any age group. One limitation of this study is the high loss to follow-up (82%). The participants lost to follow-up differed significantly in sex, physical activity level, and self-esteem at baseline. However, even though participants lost to follow-up were statistically different regarding physical activity and self-esteem, the difference was low (0.2% difference in self-esteem and 2.3% difference in low physical activity level), probably not of clinical relevance. Furthermore, we did not have data on socioeconomic or pain status at follow-up for non-responders. Nevertheless, 36% of participants lost to follow up were assumed to be missing completely at random as they either died or emigrated between the two follow-ups.

The 11-year follow-up period forces us to be careful with interpretations of the associations, as we do not have information on changes in lifestyle, education, work, health status, pain, or injuries during follow-up. This is especially relevant since the transitional stage from adolescence to young adulthood is characterised by developmental changes in the social environment, lifestyle, work situation, and final biological and psychological maturation [62–64]. Furthermore, the low number of boys compared to girls in this study could have influenced statistical power and resulted in a type II error for boys, and it precluded stratification by sex in the model building. The Samples are not mutually exclusive, including individuals with and without neck pain in Sample I. The use of non-validated, single items for loneliness, sleep, and perceived family income may have biased the associations. The question measuring physical activity level has shown moderate correlation with cardiovascular fitness, but low correlation with objectively measured total energy expenditure and physical activity level [29]. Future studies should investigate variables such as physical activity, sleep, and socioeconomic status with objective measures to provide more valid measurements. Our findings should be validated in future longitudinal studies.

## Implications

The risk profile analyses illustrated that combinations of selected risk factors in adolescence cumulatively increased the probability of developing neck pain in young adulthood. This highlights the importance of investigating combinations of risk factors to identify high-risk groups and to develop targeted prevention programs. Risk factors such as physical activity and loneliness are of special importance as these are modifiable. Our results substantiate the importance of promoting universal access to moderate and high physical activity in adolescents and motivating and facilitating adolescents who are already active to stay active. This is especially important since there is a trend towards decreased physical activity level through adolescence

and young adulthood [50]. Moreover, physical activity has the potential to reduce existing neck pain [65], and may contribute to a higher participation in teams and sporting clubs, hence increase social access to social support and prevent loneliness [66]. For health care providers, risk profiles could contribute to identifying adolescents who are most at risk of developing neck pain.

## Conclusion

In this prospective cohort study, we found that combinations of risk factors in adolescence cumulatively increased the probability of neck pain in young adulthood. Adolescents with co-occurring pain, loneliness, and inactivity are at a particularly high risk of having neck pain in young adulthood. Further, the risk is increased also for those with perceived low family income, especially girls. Targeting risk profiles in public health policy and efforts, primary health care and future intervention studies might contribute to reduce the burden of neck pain in younger populations.

## Supporting information

**S1 Table. STROBE statement—Checklist of items that should be included in reports of observational studies.**
(DOCX)

**S2 Table. Univariate analyses of the association between potential risk factors in adolescence and persistent neck pain in young adulthood.**
(DOCX)

**S3 Table. Analyses of baseline characteristics of study participants and participants lost to follow-up.**
(DOCX)

**S1 Questionnaire. HUNT4, Norwegian.**
(DOCX)

**S2 Questionnaire. HUNT4, English.**
(DOCX)

**S3 Questionnaire. Young-HUNT3, Norwegian.**
(PDF)

**S4 Questionnaire. Young-HUNT3, English.**
(PDF)

## Acknowledgments

We thank all the participants participating in the HUNT Study, which is a collaboration between HUNT Research Centre, (Faculty of Medicine and Health Sciences, NTNU, Norwegian University of Science and Technology), Trøndelag County Council, Central Norway Regional Health Authority, and the Norwegian Institute of Public Health.

## Author Contributions

**Conceptualization:** Henriette Jahre, Margreth Grotle, Milada Småstuen, Maren Hjelle Guddal, Kaja Smedbråten, Kåre Rønn Richardsen, Synne Stensland, Kjersti Storheim, Britt Elin Øiestad.

**Data curation:** Henriette Jahre, Milada Småstuen.

**Formal analysis:** Henriette Jahre, Margreth Grotle, Milada Småstuen, Maren Hjelle Guddal, Kaja Smedbråten, Kåre Rønn Richardsen, Synne Stensland, Kjersti Storheim.

**Funding acquisition:** Margreth Grotle, Britt Elin Øiestad.

**Methodology:** Henriette Jahre, Margreth Grotle, Milada Småstuen, Maren Hjelle Guddal, Kaja Smedbråten, Kåre Rønn Richardsen, Synne Stensland, Kjersti Storheim, Britt Elin Øiestad.

**Project administration:** Henriette Jahre.

**Supervision:** Margreth Grotle, Milada Småstuen, Britt Elin Øiestad.

**Visualization:** Henriette Jahre, Margreth Grotle, Milada Småstuen, Maren Hjelle Guddal, Kaja Smedbråten, Kåre Rønn Richardsen, Synne Stensland, Kjersti Storheim, Britt Elin Øiestad.

**Writing – original draft:** Henriette Jahre.

**Writing – review & editing:** Margreth Grotle, Milada Småstuen, Maren Hjelle Guddal, Kaja Smedbråten, Kåre Rønn Richardsen, Synne Stensland, Kjersti Storheim, Britt Elin Øiestad.

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
