## [Decision Letter · Decision Letter 0]

26 Apr 2021

PONE-D-20-32895

Risk factors and risk profiles for neck pain in young adults: prospective analyses from adolescence to young adulthood - The North-Trøndelag Health Study

PLOS ONE

Dear Dr. Jahre,

Thank you for submitting your manuscript to PLOS ONE. After careful consideration, we feel that it has merit but does not fully meet PLOS ONE’s publication criteria as it currently stands. Therefore, we invite you to submit a revised version of the manuscript that addresses the points raised during the review process.

We look forward to receiving your revised manuscript.

Kind regards,

Alison Rushton

Academic Editor

PLOS ONE

Journal Requirements:

2) You indicated that you had ethical approval for your study. In your Methods section, please ensure you have also stated whether you obtained consent from parents or guardians of the minors included in the study or whether the research ethics committee or IRB specifically waived the need for their consent.

3) Please include additional information regarding the survey or questionnaire used in the study and ensure that you have provided sufficient details that others could replicate the analyses. For instance, if you developed a questionnaire as part of this study and it is not under a copyright more restrictive than CC-BY, please include a copy, in both the original language and English, as Supporting Information.

4)  We note that you have indicated that data from this study are available upon request. PLOS only allows data to be available upon request if there are legal or ethical restrictions on sharing data publicly. For information on unacceptable data access restrictions, please see http://journals.plos.org/plosone/s/data-availability#loc-unacceptable-data-access-restrictions.

Reviewers' comments:

Reviewer's Responses to Questions

**Comments to the Author**

1. Is the manuscript technically sound, and do the data support the conclusions?

Reviewer #1: Yes

Reviewer #2: Yes

Reviewer #3: Partly

2. Has the statistical analysis been performed appropriately and rigorously? 

Reviewer #1: Yes

Reviewer #2: Yes

Reviewer #3: Yes

3. Have the authors made all data underlying the findings in their manuscript fully available?

Reviewer #1: Yes

Reviewer #2: No

Reviewer #3: Yes

4. Is the manuscript presented in an intelligible fashion and written in standard English?

Reviewer #1: Yes

Reviewer #2: Yes

Reviewer #3: No

5. Review Comments to the Author

Reviewer #1: 

The manuscript “Risk factors and risk profiles for neck pain in young adults: prospective analyses from adolescence to young adulthood - The North-Trøndelag Health Study”. As a result of this study, The risk profiles in both samples showed that co-occurrence of risk factors, such as headache/migraine, neck/shoulder pain, back pain,low physical activity level, loneliness, and low family income cumulatively increased the probability of neck pain in young adulthood. Although the study is relevant and meticulously crafted, there are four issues that needs addressing by authors before publication.

* In the introduction, the emphasis on the originality of the study is not properly made. this issue should be underlined before the purpose

* In the discussion part, cultural differences in comparisons with other studies should be highlighted. Are these studies done with similar cultures?

*The reason for the low physical activity or the effect of this lowness on muscle strength and its relation with the neck muscles can be examined. Could the increase in internet usage be a risk factor in this regard?

*İt can increase internet usage in loneliness. Can this study be compared and discussed with the studies that have been done in the past when there was no internet use?

Reviewer #2: 

I have had the privilege of reviewing the manuscript entitled: Risk factors and risk profiles for neck pain in young adults: prospective analyses from adolescence to young adulthood - The North-Trøndelag Health Study. The authors set out to investigate risk factors for neck pain in adolescents and young adults using a prospective design. This is a very good manuscript and I would like to start off with congratulating the authors for that. It is clear, well written and to-the-point.

I do have some comments that I would like the authors to address before the manuscript is ready for publication.

P1 L56-57. I’m not sure I follow the argument that because previous studies have found that daytime tiredness and use of text messages are risk factors for neck pain, that you should investigate it further? Can you please elaborate more on this?

P7 L194. I imagine that the perceived family income will be highly affected by the area in which the participants live and the people they are mostly around. I this a validated question?

P7. I like your thorough explanations of all your exposures and outcomes. Very nice!

P11 L271. S2 Table is a table of comparison between responders and non-responders and not the univariate analyses. Please correct.

P13 L315-316. Please use “Perceived family income” instead of “Family income” throughout the manuscript.

P14 L389-391. I think this statement is somewhat far-fetched. What mechanisms in the muscles are you referring to? Please justify this with more that one reference in your local language.

P15 L361. Many of your exposures are self-reported even though some could have been quantified e.g. physical activity, family income etc. Please elaborate on the implications and arguments of this choice.

P17 L415. Please elaborate more on the differences between the responders and non-responders. As stated in line 242 the non-responders have higher level of physical activity and higher self-esteem. This would push the group towards less pain if the non-responders were included. I think this is a crucial part of your study, so I would expect you to have an in-depth discussion about it.

Reviewer #3: 

GENERAL COMMENT

Thank you for the invitation to review this paper. The paper has evaluated a critical aspect of possible risk factors for neck pain in adolescents. Although it has an apropriate methodology and interesting reuslts some critical issues were identified. Introduction, Discussion and Limitations presented with some critical weaknesses that need authors attention if a re-submission will be considered. Authors should make sure that the manuscript is read and corrected by a native English- speaker. This is very important to ensure that the presenation and key messages are clear. Several comments highlight difficulties in the way that the manuscript is preseneted.

Introduction

Line 50: Authors refer to studies, however, the reference at the end of the sentence includes one relative old study about back pain in adolescents. Further justification of the statement must be included.

line 52-59: Authors description decreases the strength of the statement. The paragraph ends with the need of the research question; still, needs a better flow. Please, rephrase.

Materials and Methods

Line 97: Was this the only exclusion criteria used? If not, please, provide all exlusion/inclusion criteria of the study.

Lines 143-144: is this a post-hoc analysis? If yes, please state it.

Lines 155-156 Please provide results of the acceptable validity and reliability in parenthesis

It seems that the current questionnaire has a substantial reliability only for girls while based on the original study especially among girls. None of the questionnaires however seemed to be a valid instrument for measuring physical activity compared to TEE and PAL in adolescents. This should be discussed as a limitation in the study.

Line 173 Other study or studies? Please be precise when justifying a statement for the outcome measures of the study.

Results

Table 1: Abbreviations of BMI, yr, SD etc. are missing

Discussion

Line 332: what type of pain? neck pain?

Line 333: which unhelathy variables?

Line 337: Can you identify which factor may play a more critical role? This could make a difference

Line 359-360: Where this assumption comes from? Please, explain and justify

Line 362: Refernces should be placed after the comma

Line 367: Can you justify this assumption?

Lines 356-370 You should use a separate sentence as this section because it is difficult to read.

Lines 390 Why muscles are affected? Do you have such indication/measurement from the results of the present study. This statement is very debatable for pain neuroscience. Please, re-consider it

Line 394 Which are these studies?Please use references.

Line 401 Did this factor affected your Power analysis?

Lines 405-408 paragraph should be rephrased. It is vague and difficult to digest.

Line 411-427

Limitations: Have you considered spinal deformities as a possible risk factor? During the 11 years several other factors like whiplash injuries, workload etc. may have changed the presence or recurrence neck pain. Could have these factors been systematic errors affecting results?

Line 417 How much losses to-follow-up ?

Lines 421-422 This issue should be discussed further in discussion and compared to other studies.

Conclusion

Lines 433-436 Are these factors associated? If physical activity is high other factors are less critical? Have you considered if parental socioeconomic status affects adolescents participation in physical activity due to motivation etc.?

6. PLOS authors have the option to publish the peer review history of their article (what does this mean?). If published, this will include your full peer review and any attached files.

Reviewer #1: **Yes: **Nuray ALACA

Reviewer #2: **Yes: **Dr. Henrik Koblauch

Reviewer #3: **Yes: **Stefanos Karanasios

---

## [Author Response · Author response to Decision Letter 0]

27 May 2021

Journal Requirements:

Author response and action: We thank the editor for highlighting the importance of following the journals guidelines. We have now thoroughly gone through the journal’s requirements and edited the manuscript accordingly. 

 2) You indicated that you had ethical approval for your study. In your Methods section, please ensure you have also stated whether you obtained consent from parents or guardians of the minors included in the study or whether the research ethics committee or IRB specifically waived the need for their consent.

Author response: 

We thank editor for this comment. We agree that this is important information that should be included in the method section. We have provided this information. 

Author action: Page 2, Line number 85-87: Participation in the HUNT study was voluntary, and all study participants signed a written consent form. Written consent from a guardian was required for participants under the age of 16 years.

3) Please include additional information regarding the survey or questionnaire used in the study and ensure that you have provided sufficient details that others could replicate the analyses. For instance, if you developed a questionnaire as part of this study and it is not under a copyright more restrictive than CC-BY, please include a copy, in both the original language and English, as Supporting Information.

Author response and action: Thank you for this comment. The questionnaires used in this study was in Norwegian. The HUNT Research Centre has developed English questionnaires to inform foreign researchers about the study methods, but these are not validated for research in English. We have provided these questionnaires used in both English and Norwegian as Supporting Information. 

 4) We note that you have indicated that data from this study are available upon request. PLOS only allows data to be available upon request if there are legal or ethical restrictions on sharing data publicly. For information on unacceptable data access restrictions, please see http://journals.plos.org/plosone/s/data-availability#loc-unacceptable-data-access-restrictions.

Author response: We acknowledge that data sharing is important. Unfortunately, due to restrictions from the Regional Committee for Medical and Health Research Ethics in accordance with Norwegian law, the participants in the HUNT study have not given consent to public sharing of their data. Therefore, data are only available upon request to the Data Access Committee at the HUNT Research Centre. We included this information in the cover letter.

Author action: Data cannot be shared publicly because of restrictions from the Regional Committees for Medical and Health Research Ethics (post@helseforskning.etikkom.no) in

accordance with Norwegian law, as participants in the HUNT survey have not given

consent to public sharing of their data. Therefore, these data are only available upon request to the Data Access Committee at HUNT Research Centre at hunt@medisin.ntnu.no.

Reviewer #1: 

The manuscript “Risk factors and risk profiles for neck pain in young adults: prospective analyses from adolescence to young adulthood - The North-Trøndelag Health Study”. As a result of this study, The risk profiles in both samples showed that co-occurrence of risk factors, such as headache/migraine, neck/shoulder pain, back pain, low physical activity level, loneliness, and low family income cumulatively increased the probability of neck pain in young adulthood. Although the study is relevant and meticulously crafted, there are four issues that needs addressing by authors before publication.

 1) In the introduction, the emphasis on the originality of the study is not properly made. this issue should be underlined before the purpose

Author comments: We are thankful to the reviewer for highlighting the need to make a clearer presentation of the study’s originality to the introduction. We agree with the reviewer and have made changes in the introduction. 

Author actions: Page 1 and 2, Line number 67-72: Identifying single risk factors and the co-occurrence of risk factors in adolescents will enable us to identify high-risk groups. Such knowledge will contribute to designing future preventive interventions with the aim of reducing the high burden of neck pain for both the society and the individuals affected. To the best of our knowledge, no previous studies have investigated co-occurrence of risk factors (risk profiles) for neck pain in adolescents.

 2) In the discussion part, cultural differences in comparisons with other studies should be highlighted. Are these studies done with similar cultures?

Author response: Thank you for this suggestion. We agree with the reviewer that cultural differences might play an important role when comparing results across studies. We provided information of which countries the comparable studies came from and added a sentence about cultural differences regarding the socioeconomic perspective in the discussion.

Author action:

Page 14, Line number 334: Norwegian adolescents

Page 15, Line number 373: Scandinavian studies

Page 16, Line number 389-393: It is essential to consider the cultural aspect of this question. In a country like Norway with a high welfare standard, perceived low family income may be more related to possibilities for social participation rather than poverty related to having access to water, food, or healthcare. Thus, our results might be more related to the social aspects of having a low family income, such as lower possibilities of social participation in different activities.

3) The reason for the low physical activity or the effect of this lowness on muscle strength and its relation with the neck muscles can be examined. Could the increase in internet usage be a risk factor in this regard?

Author comment: We understand this comment is related to the discussion on physical activity. We agree that internet use could be associated with decreased muscle strength and thus also neck muscles, and in that way may be a potential risk factor for neck pain. However, we do not have access to internet usage in this sample. In addition, internet usage among adolescents has probably changed considerably since 2006-2008, and data from that time point may not be relevant for today’s population regarding internet use. 

4) İt can increase internet usage in loneliness. Can this study be compared and discussed with the studies that have been done in the past when there was no internet use?

 Author comment: Thank you for highlighting the influence of internet. Because baseline data was measured in 2006-2008, we will be careful to discuss internet use as this has changed enormously the last decade. Further, there are not many studies investigating loneliness in relationship with neck pain in adolescents, so comparison with more studies from the past is difficult. 

Reviewer #2: 

I have had the privilege of reviewing the manuscript entitled: Risk factors and risk profiles for neck pain in young adults: prospective analyses from adolescence to young adulthood - The North-Trøndelag Health Study. The authors set out to investigate risk factors for neck pain in adolescents and young adults using a prospective design. This is a very good manuscript and I would like to start off with congratulating the authors for that. It is clear, well written and to-the-point.

I do have some comments that I would like the authors to address before the manuscript is ready for publication.

 1) P1 L56-57. I’m not sure I follow the argument that because previous studies have found that daytime tiredness and use of text messages are risk factors for neck pain, that you should investigate it further? Can you please elaborate more on this?

Author response and actions: We are thankful to the reviewer for highlighting the need for making this paragraph clearer. We agree that our argument of why it is important to investigate neck pain is not clear. After further consideration, we decided to remove the paragraph, as we do not think it adds something necessary to the introduction. 

 2) P7 L194. I imagine that the perceived family income will be highly affected by the area in which the participants live and the people they are mostly around. Is this a validated question?

Author response: Thank you for highlighting the need to specify this. We agree with the reviewer that perceived family income probably is affected by the living area. We provided a sentence regarding validation in the method section and added a sentence regarding cultural differences in the discussion section. 

Author actions: 

Page 7, Line number 196-197: This question has shown correspondence with parents’ education and parents’ work affiliation in a previous Norwegian study (39), but is not formally validated

Page 16, Line number 389-393: It is essential to consider the cultural aspect of this question. In a country like Norway with a high welfare standard, perceived low family income may be more related to possibilities for social participation rather than poverty related to having access to water, food, or healthcare. Thus, our results might be more related to the social aspects of having a low family income, such as lower possibilities of social participation in different activities.

 3) P7. I like your thorough explanations of all your exposures and outcomes. Very nice!

Author response: We thank the reviewer for this positive comment! 

 4) P11 L271. S2 Table is a table of comparison between responders and non-responders and not the univariate analyses. Please correct.

Author response: Thank you for pointing out this important mistake.

Author action: We replaced the table of responders and non-responders with the univariate analyses.

 5) P13 L315-316. Please use “Perceived family income” instead of “Family income” throughout the manuscript.

Author response: Thank you for this comment. We agree with the reviewer. 

Author action: We changed “family income” to “perceived family income” throughout the manuscript.

 6) P14 L389-391. I think this statement is somewhat far-fetched. What mechanisms in the muscles are you referring to? Please justify this with more that one reference in your local language.

Author response and actions: Thank you for highlighting that this statement should be justified better. We agree that this is not well documented. Since reviewer three also questioned this sentence and recommended we reconsider it, we decided to remove it.

 7) P15 L361. Many of your exposures are self-reported even though some could have been quantified e.g. physical activity, family income etc. Please elaborate on the implications and arguments of this choice.

Author response: We agree that more objectively measures of these variables could provide more valid results. Since we used already collected data from the HUNT study, we had to use the available variables. We included a sentence under the limitation section to highlight this issue.

Author action: Page 18, Line number 437-439: Future studies should investigate variables such as physical activity, sleep, and socioeconomic status with objective measures to provide more valid measurements.

 8) P17 L415. Please elaborate more on the differences between the responders and non-responders. As stated in line 242 the non-responders have higher level of physical activity and higher self-esteem. This would push the group towards less pain if the non-responders were included. I think this is a crucial part of your study, so I would expect you to have an in-depth discussion about it.

Author response: Thank you for highlighting this important aspect. When conducting the statistical analyses of differences between responders and participants lost to follow-up, sex, physical activity, and self-esteem were statistically significantly different. However, these differences were small regarding physical activity level (2.3% difference) and self-esteem (0.2% difference), so we assume that this is not of any clinical relevance or have affected the results. However, we agree with the reviewer that this should be clearly stated in the manuscript. We included a sentence under the strength and limitation section.

Author actions: Page 17, Line number 418-421: However, even though participants lost to follow-up were statistically different regarding physical activity and self-esteem, the difference was low (0.2 % difference in self-esteem and 2.3% difference in low physical activity level), probably not of clinical relevance.

Reviewer #3: 

GENERAL COMMENT

Thank you for the invitation to review this paper. The paper has evaluated a critical aspect of possible risk factors for neck pain in adolescents. Although it has an appropriate methodology and interesting results some critical issues were identified. Introduction, Discussion and Limitations presented with some critical weaknesses that need authors attention if a re-submission will be considered. Authors should make sure that the manuscript is read and corrected by a native English- speaker. This is very important to ensure that the presentation and key messages are clear. Several comments highlight difficulties in the way that the manuscript is presented.

Introduction

 1) Line 50: Authors refer to studies, however, the reference at the end of the sentence includes one relative old study about back pain in adolescents. Further justification of the statement must be included.

Author response: We thank the reviewer for highlighting this issue; we agree that this sentence need further justification. We included two more references of newer date after this statement. 

Author actions: Page 1, Line 54: We included two more references (7 and 8)

 2) line 52-59: Authors description decreases the strength of the statement. The paragraph ends with the need of the research question; still, needs a better flow. Please, rephrase.

Author response and actions: We are thankful to the reviewer for highlighting the need for making this paragraph clearer. We agree that our argument of why it is important to investigate neck pain is not clear in this section. After further consideration, we decided to remove the paragraph, as we do not think it adds something necessary to the introduction.

Materials and Methods

 3) Line 97: Was this the only exclusion criteria used? If not, please, provide all exlusion/inclusion criteria of the study.

Author response: Thank you for this question. Yes, adolescents with juvenile arthritis were the only ones excluded from inclusion at baseline. 

4) Lines 143-144: is this a post-hoc analysis? If yes, please state it.

Author response: Thank you for this question. No, this was not a post-hoc analysis. 

This is how we categorized the variable, but the categorization was conducted after receiving the data. 

 5) Lines 155-156 Please provide results of the acceptable validity and reliability in parenthesis

It seems that the current questionnaire has a substantial reliability only for girls while based on the original study especially among girls. None of the questionnaires however seemed to be a valid instrument for measuring physical activity compared to TEE and PAL in adolescents. This should be discussed as a limitation in the study.

Author response: Thank you for highlighting the need for elaboration of this question’s weaknesses. We agree that this is highly relevant information and have provided additional information in the method section and in the discussion section.

Author actions: Page 5, Line 154-155: Physical activity was assessed with a question adapted from the World Health Organization Health Behaviour in Schoolchildren (HBSC) study (28). The question has showed to correlate with cardiovascular fitness (r=0.39), especially for girls (r=0.55) 

Page 18, Line number 434-437: The question measuring physical activity level has shown moderate correlation with cardiovascular fitness, but low correlation with objectively measured total energy expenditure and physical activity level

 6) Line 173 Other study or studies? Please be precise when justifying a statement for the outcome measures of the study.

Author response: Thank you for highlighting the importance of being precise; we changed to one study. 

Author actions: Page 6, Line number 172: ….as suggested in one study

Results

 7) Table 1: Abbreviations of BMI, yr, SD etc. are missing

Author response: Thank you for pointing this out. We agree that this should be included. 

Author actions: We provided the abbreviations for yr, year; SD, standard deviation; BMI, Body mass index in the table legend.

Discussion

 8) Line 332: what type of pain? neck pain?

Author response: This study investigated chronic non-specific pain. We specified this.

Author action: Page 14, Line number 333-334: ….non-specific pain in Norwegian adolescents

 9) Line 333: which unhealthy variables?

Author response: Thank you for this comment. We added the unhealthy variables included in the study.

Author action: Page 14, Line number 334-336: They found a gradually stronger association with a higher number of unhealthy variables (low physical activity level, sedentary behaviour, high BMI, smoking, and alcohol). 

 10) Line 337: Can you identify which factor may play a more critical role? This could make a difference

Author response: Thank you for this question. We agree that this would be interesting to know. As seen in the risk matrix, female sex is the only variable in all the combinations with the highest probability of neck pain. Except for this, it seems like an increasing number of “risk factors” influence the probability of neck pain regardless of type. We have specified this in the discussion.

Author actions: Page 14, Line number 340-341: Girls had, in general, a higher probability of neck pain than boys regardless of the combinations of risk factors.

 11) Line 359-360: Where this assumption comes from? Please, explain and justify

Author response: We agree that we should elaborate on this. We rewrote this sentence to make it clearer.

Author actions: Page 15, Line number 363-364: Our finding of a high prevalence of neck/shoulder pain in adolescence and the impact early pain had on future pain indicate that pain develops early.

 12) Line 362: References should be placed after the comma

Author response: Thank you for pointing this out. We agree and placed the references earlier in the sentence.

Author action: Page 15, Line number 365: Contrary to previous studies (12, 20), we found that a low level of physical activity in adolescence was associated with neck pain in young adulthood (Sample I).

 13) Line 367: Can you justify this assumption?

Author response: Thank you for this question. This sentence is speculation, and we do not have strong evidence to support this. Since we cannot justify this assumption, we decided to remove the sentence.

Author action: We removed the sentence,

 14) Lines 356-370 You should use a separate sentence as this section because it is difficult to read.

Author response: Thank you for your comment, we agree with the reviewer that this paragraph is difficult to read. We rephrased the paragraph.

Author actions: Page 15, Line number 357-362: The reasons for these associations are unclear, but changes caused by pain in one body site might influence other body sites, and share similar mechanisms. One explanation for pain in different body parts is the neurophysiological changes implicated in central sensitisation. Further, studies have shown alterations in pain processing after an episode of acute pain, which seems to influence pain persistence in adolescents

 15) Lines 390 Why muscles are affected? Do you have such indication/measurement from the results of the present study. This statement is very debatable for pain neuroscience. Please, re-consider it

Author response: Thank you for highlighting that this is very debatable for pain neuroscience. We do not have measurement from this study to support this assumption. We decided to remove this sentence from the manuscript. 

Author action: We removed the sentence. 

 16) Line 394 Which are these studies? Please use references.

Author response: Thank you for this question and for highlighting the need for more references. We agree with the reviewer and have provided more references and rewritten the sentence. 

Author action: Page 16, Line number 396-400: An explanation for these results can be that Sample II excluded those with neck/shoulder pain at baseline. Previous studies have shown that adolescents experiencing pain have other illness perceptions or health behaviours than those who are pain-free. This includes withdrawal from social (57, 58) and pain-provoking physical activities (3, 58), reduced sleep quality (58, 59), decreased quality of life (4), and lower psychosocial well-being (2). These factors might influence future pain experience (60).

 17) Line 401 Did this factor affected your Power analysis?

Author response: Thank you for this question. Yes, a reduction in statistical power could be an explanation for our results. We included a sentence about this. 

Author action: Page 16, Line number 400-401: Another reason for different results between the two samples might be lack of statistical power due to the sample sizes (1422 vs 832).

 18) Lines 405-408 paragraph should be rephrased. It is vague and difficult to digest.

Author response: We agree that this paragraph was vague, so we rephrased it.

Author action: Page 17, Line number 406-409: Psychological distress and sleeping difficulties did not reach statistical significance in our multiple analyses. This is contrary to findings from previous studies (23, 61). Potential explanations may be different measurements used, different follow-up periods, and different statistical models. 

 19) Line 411-427

Limitations: Have you considered spinal deformities as a possible risk factor? During the 11 years several other factors like whiplash injuries, workload etc. may have changed the presence or recurrence neck pain. Could have these factors been systematic errors affecting results?

Author response: Thank you for this question. We acknowledge that there probably are other risk factors for neck pain that is not covered in this study. However, to the best of our knowledge, spinal deformities have not shown any previous association with neck pain in longitudinal studies of adolescents or young adults. There is, indeed, an important limitation that we do not have data during the 11-year follow-up. Thank you for pointing this out. We have now elaborated on this in the limitation section.

Author action: Page 18, Line number 425-430: The 11-year follow-up period forces us to be careful with interpretations of the associations, as we do not have information on changes in lifestyle, education, work, health status, pain, or injuries during follow-up. This is especially relevant since the transitional stage from adolescence to young adulthood is characterised by developmental changes in the social environment, lifestyle, work situation, and final biological and psychological maturation (62-64). 

 20) Line 417 How much losses to-follow-up ?

Author response: We agree that this is relevant information and provided the percent of loss to follow-up.

Author action: Page 17, Line number: 416-417 One limitation of this study is the high loss to follow-up (82%).

 21) Lines 421-422 This issue should be discussed further in discussion and compared to other studies.

Author response: Thank you for this comment. We agree that this is important to discuss. We discussed this further in the limitation section. 

Author action: Page 18, Line number 425-430: The 11-year follow-up period forces us to be careful with interpretations of the associations, as we do not have information on changes in lifestyle, education, work, health status, pain, or injuries during follow-up. This is especially relevant since the transitional stage from adolescence to young adulthood is characterised by developmental changes in the social environment, lifestyle, work situation, and final biological and psychological maturation (62-64). 

Conclusion

 22) Lines 433-436 Are these factors associated? If physical activity is high other factors are less critical? Have you considered if parental socioeconomic status affects adolescents participation in physical activity due to motivation etc.?

Author response: Thank you for these questions. We chose to have extra focus on the implications regarding physical activity as this is a modifiable variable and also have the potential to decrease neck pain and loneliness. We rephrased the sentence to make this more clear. The association between socioeconomic status and low physical activity level is not investigated in this study, and to the best of our knowledge, the evidence regarding this association is unclear in adolescents. We remove this from the paragraph. 

Author action: Page 18-19, Line number 445-450: Risk factors such as physical activity and loneliness are of special importance as these are modifiable. Our results substantiate the importance of promoting universal access to moderate and high physical activity in adolescents and motivating and facilitating adolescents who are already active to stay active. This is especially important since there is a trend towards decreased physical activity level through adolescence and young adulthood.

---

## [Decision Letter · Decision Letter 1]

22 Jul 2021

PONE-D-20-32895R1

Risk factors and risk profiles for neck pain in young adults: prospective analyses from adolescence to young adulthood - The North-Trøndelag Health Study

PLOS ONE

Dear Dr. Jahre,

Thank you for submitting your manuscript to PLOS ONE. After careful consideration, we feel that it has merit but does not fully meet PLOS ONE’s publication criteria as it currently stands. Therefore, we invite you to submit a revised version of the manuscript that addresses the points raised during the review process.

Please provide the last additions (references and notes) requested by reviewer n°3. 

We look forward to receiving your revised manuscript.

Kind regards,

Andrea Martinuzzi

Academic Editor

PLOS ONE

Journal Requirements:

Reviewers' comments:

Reviewer's Responses to Questions

**Comments to the Author**

1. If the authors have adequately addressed your comments raised in a previous round of review and you feel that this manuscript is now acceptable for publication, you may indicate that here to bypass the “Comments to the Author” section, enter your conflict of interest statement in the “Confidential to Editor” section, and submit your "Accept" recommendation.

Reviewer #1: All comments have been addressed

Reviewer #2: All comments have been addressed

Reviewer #3: All comments have been addressed

2. Is the manuscript technically sound, and do the data support the conclusions?

Reviewer #1: No

Reviewer #2: Yes

Reviewer #3: Yes

3. Has the statistical analysis been performed appropriately and rigorously? 

Reviewer #1: No

Reviewer #2: Yes

Reviewer #3: Yes

4. Have the authors made all data underlying the findings in their manuscript fully available?

Reviewer #1: Yes

Reviewer #2: No

Reviewer #3: Yes

5. Is the manuscript presented in an intelligible fashion and written in standard English?

Reviewer #1: Yes

Reviewer #2: Yes

Reviewer #3: Yes

6. Review Comments to the Author

Reviewer #1: (No Response)

Reviewer #2: (No Response)

Reviewer #3: Authors responded adequately to all reviewers suggestions and made essential improvements with their revised manuscript. Some minor changes should be made as follows

Line 52 Please provide a reference for the following statement if exists ‘The high prevalence of neck pain in adolescents …’

Line 224-228 Please provide a reference to justify the formula used

Line 326-330 Please specify that your findings are describing an association in adolescents in Norway

Line 355-356 Similar as above comment

Subsequently, a final decicion for approval should be supported

7. PLOS authors have the option to publish the peer review history of their article (what does this mean?). If published, this will include your full peer review and any attached files.

Reviewer #1: **Yes: **Nuray ALACA

Reviewer #2: **Yes: **Henrik Koblauch

Reviewer #3: **Yes: **Stefanos Karanasios

---

## [Author Response · Author response to Decision Letter 1]

28 Jul 2021

Author comment: Thank you for the constructive comments. The suggested changes are conducted as described below.

1) Line 52 Please provide a reference for the following statement if exists ‘The high prevalence of neck pain in adolescents …’

Author comment: Thank you for this comment, we agree that this statement should be supported by references. 

Author action: 

Page 1, Line 52: We provided two references to support the statement that neck pain is prevalent in adolescents. 

2) Line 224-228 Please provide a reference to justify the formula used

Author comment: Thank you for pointing this out.

Author action: 

Page 8, Line 225: We inserted a references to justify the formula used. 

3) Line 326-330 Please specify that your findings are describing an association in adolescents in Norway

Author comment: Thank you for highlighting this, we rewrote the sentence. 

Author changes: 

Page 13, Line 327: This study found that female sex, low level of physical activity, loneliness, headache/migraine, back pain, and neck/shoulder pain in adolescence were risk factors for having neck pain in young adulthood in Norwegian adolescents.

4) Line 355-356 Similar as above comment

Author comment: Thank you for highlighting this, we agree that this should be specified. 

Author actions: 

Page 15, Line 357: We rewrote the sentence: Our finding that headache/migraine, neck/shoulder, and back pain in adolescence were associated with neck pain in young adulthood among Norwegian adolescents is supported in the literature

---

## [Editor Report · Decision Letter 2]

29 Jul 2021

Risk factors and risk profiles for neck pain in young adults: prospective analyses from adolescence to young adulthood - The North-Trøndelag Health Study

PONE-D-20-32895R2

Dear Dr. Jahre,

We’re pleased to inform you that your manuscript has been judged scientifically suitable for publication and will be formally accepted for publication once it meets all outstanding technical requirements.

Kind regards,

Andrea Martinuzzi

Academic Editor

PLOS ONE
---

## [Editor Report · Acceptance letter]

2 Aug 2021

PONE-D-20-32895R2 

Risk factors and risk profiles for neck pain in young adults: prospective analyses from adolescence to young adulthood - The North-Trøndelag Health Study 

Dear Dr. Jahre:

I'm pleased to inform you that your manuscript has been deemed suitable for publication in PLOS ONE. Congratulations! Your manuscript is now with our production department. 

Kind regards, 

on behalf of

Dr. Andrea Martinuzzi 

Academic Editor

PLOS ONE